# Local and Systemic Adverse Reactions to mRNA COVID-19 Vaccines Comparing Two Vaccine Types and Occurrence of Previous COVID-19 Infection

**DOI:** 10.3390/vaccines9121463

**Published:** 2021-12-10

**Authors:** Oleguer Parés-Badell, Xavier Martínez-Gómez, Laia Pinós, Blanca Borras-Bermejo, Sonia Uriona, Susana Otero-Romero, José Ángel Rodrigo-Pendás, Yolima Cossio-Gil, Antònia Agustí, Cristina Aguilera, Magda Campins

**Affiliations:** 1Servei de Medicina Preventiva i Epidemiologia, Vall d’Hebron Hospital Universitari, Vall d’Hebron Barcelona Hospital Campus, Passeig Vall d’Hebron 119-129, 08035 Barcelona, Spain; opares@vhebron.net (O.P.-B.); lpinos@vhebron.net (L.P.); bborras@vhebron.net (B.B.-B.); smuriona@vhebron.net (S.U.); sotero@vhebron.net (S.O.-R.); jarodrigo@vhebron.net (J.Á.R.-P.); mcampins@vhebron.net (M.C.); 2Department of Experimental Sciences and Health, Universitat Pompeu Fabra, 08005 Barcelona, Spain; 3Unitat Docent Vall d’Hebron, Universitat Autònoma de Barcelona, 08193 Bellaterra, Spain; 4Department Information Systems and Decision Support, Hospital Universitari Vall d’Hebron, 08035 Barcelona, Spain; ycossio@vhebron.net; 5Research Group of Healthcare System Management, Vall d’Hebron Institut de Recerca (VHIR), 08035 Barcelona, Spain; 6Servei de Farmacologia Clínica, Vall d’Hebron Institut de Recerca (VHIR), 08035 Barcelona, Spain; aagusti@vhebron.net (A.A.); craguile@vhebron.net (C.A.); 7Departament de Farmacologia, Terapèutica i Toxicologia, Universitat Autònoma de Barcelona, 08193 Bellaterra, Spain

**Keywords:** SARS-CoV-2, COVID-19-vaccination, mRNA vaccines, adverse reactions, healthcare workers

## Abstract

The aim of this study was to assess adverse reactions to COVID-19 vaccines, comparing the BNT162b2 or the mRNA-1273 COVID-19 vaccines and the presence and seriousness of a previous COVID-19 infection. We conducted a cross-sectional online survey of vaccinated healthcare workers at a tertiary hospital in Barcelona (Spain). Thirty-eight percent of vaccine recipients responded to the questionnaire. We compared the prevalence of adverse reactions by vaccine type and history of COVID-19 infections. A total of 2373 respondents had received the BNT162b2 vaccine, and 506 the mRNA-1273 vaccine. The prevalence of at least one adverse reaction with doses 1 and 2 was 41% and 70%, respectively, in the BNT162b2 group, and 60% and 92% in the mRNA-1273 group (*p* < 0.001). The BNT162b2 group reported less prevalence of all adverse reactions. Need for medical leave was significantly more frequent among the mRNA-1273 group (12% versus 4.6% *p* < 0.001). Interestingly, respondents with a history of allergies or chronic illnesses did not report more adverse reactions. The frequency of adverse reactions with dose 2 was 96% (95% CI 88–100%) for those with a history of COVID-19 related hospitalization, and 86% (95% CI 83–89%) for those with mild or moderate symptomatic COVID-19, significantly higher than for participants with no history of COVID-19 infections (67%, 95% CI 65–69%). Our results could help inform vaccine recipients of the probability of their having adverse reactions to COVID-19 vaccines.

## 1. Introduction

The World Health Organization characterized COVID-19 as a pandemic on 11 March 2020 [1]. Healthcare workers have experienced a significant burden of the disease throughout the COVID-19 pandemic [2,3]. According to the ECDC (European Centre for Disease Prevention and Control), at the beginning of the pandemic between 9% and 26% of all diagnosed COVID-19 cases in Europe were amongst healthcare workers [4].

After the initial implementation of hospital-based infection control policies and procedures, the authorization of two messenger RNA (mRNA) vaccines at the beginning of 2021—the BNT162b2 Comirnaty vaccine (Pfizer-BioNTech) [5,6] and the mRNA-1273 Spikevax vaccine (Moderna) [7,8]—was a critical event in the response to the pandemic.

Both vaccines were shown in clinical trials [5,7] to be very effective at preventing severe disease and hospitalization and to have favorable safety profiles. The European Medicines Agency recommended the emergency use authorization of the BNT162b2 vaccine on 21 December 2020 [9] and the mRNA-1273 vaccine on 6 January 2021 [10]. Spain began the COVID-19 vaccination program in December 2020, prioritizing healthcare workers and staff and residents of long-term care facilities [11].

The frequency and severity of adverse reactions to COVID-19 vaccines may vary depending on population characteristics or the vaccine type. Therefore, it is important to continue to monitor the occurrence of adverse reactions after vaccines are authorized. We conducted an online self-report survey to assess and compare the prevalence and characteristics of adverse reactions after two doses of the BNT162b2 or the mRNA-1273 COVID-19 vaccines were administered. Furthermore, we aimed to explore whether a previous COVID-19 infection and its seriousness could influence the prevalence of adverse reactions to COVID-19 vaccines.

## 2. Materials and Methods

All healthcare workers at a tertiary hospital in Barcelona (Hospital Universitari Vall d’Hebron) were invited to receive the COVID-19 vaccination. The hospital serves a population of 1.2 million people and has more than 7000 health professionals. The vaccination campaign started on 4 January 2021 with the BNT162b2 vaccine, prioritizing healthcare workers with direct contact with COVID-19 patients. The administration of the first doses of the mRNA-1273 vaccine started on 4 February 2021. Both vaccines were used depending on their availability and administered according to the vaccination protocols.

This is a cross-sectional study using an online ad hoc survey. All vaccinated healthcare workers received a mobile text message inviting them to answer a self-reported questionnaire that was available from 10 February 2021 to 3 May 2021. Those vaccinated after launching the survey received the invitation to participate 5 days after the second vaccine dose. The questionnaire was also available through the institutional webpage.

The questionnaire (Appendix A) collected information on age, gender, worker category, history of severe allergic reaction, history of chronic illness, history and seriousness of COVID-19 infection, vaccine type, dates of vaccination, adverse reaction to the first dose, adverse reaction to the second dose, onset and end of the adverse reactions, need for medical attention, need for medical leave, and potential life-threatening reactions. History of severe allergic reaction was defined as having suffered an anaphylactic shock or glottis oedema [12]. We adapted the Center for Disease Control and Prevention definition of chronic illnesses [13] to include cardiac insufficiency, ischemic heart disease, asthma, diabetes, chronic bronchitis, neurological disease, kidney failure, or chronic liver disease. Voluntarily, participants could reveal their health record’s ID code, which allowed us to review self-reported severe reactions in participants’ clinical histories.

We calculated the prevalence and 95% confidence intervals of adverse reactions to COVID-19 vaccination. The prevalence of adverse reactions was compared between the BNT162b2 and the mRNA-1273 vaccines, the first and second dose, and the history and seriousness of a previous COVID-19 infection. Categorical variables were compared using Pearson’s chi-squared test and Fisher’s exact test. Continuous variables were compared using Wilcoxon or Kruskal–Wallis rank sum tests. Data were analysed using the statistical computing program R. The study was approved by the Vall d’Hebron Ethics Committee (approval code: PR(SC)103/2021).

## 3. Results

A total of 7695 healthcare workers at our centre received a COVID-19 vaccine between 4 January 2021 and 3 May 2021. A total of 2929 of those (38.0%) responded to the self-reported questionnaire (Table 1). A total of 2373 participants received the BNT162b2 vaccine, 506 participants received the mRNA-1273 vaccine, and 50 participants did not report the vaccine type. Around 80% of the participants were women and 77% of those were between 18 and 55 years old. A total of 817 participants (27.9%) reported a history of previous COVID-19 infection. Among them, 654 (80%) had had mild to moderate symptoms and 32 (3.9%) had been hospitalized. Among those vaccinated with the BNT162b2 vaccine, 635 (26.8%) reported a history of previous COVID-19 infection. Among those vaccinated with the mRNA-1273 vaccine, 168 (33.2%) reported a history of previous COVID-19 infection.

In the BNT162b2 group, 40.6% (95% CI 38.6–42.6%) reported at least one adverse reaction with the first dose and 70.2% (95% CI 68.4–72.1%) with the second dose (Table 2). In the mRNA-1273 group, self-reporting of any adverse reaction reached 59.9% (95% CI 55.6–64.2%) with the first dose and 91.7% (95% CI 86.8–96.6%) with the second dose. All adverse reactions were more frequently reported by the mRNA-1273 vaccinated group compared to the BNT162b2 group. Pain at the injection site was the most common local reaction (Appendix A). Fatigue, headache and malaise were the most common systemic reactions for both vaccines and doses. However, the BNT162b2 vaccinated group had significantly lower frequencies than the mRNA-1273 vaccinated group. In the BNT162b2 group, 10.8% (95% CI 9.5–12.0%) reported needing a medical leave after the second dose, while in the mRNA-1273 group it was 25.6% (95% CI 17.8–33.4%, *p* < 0.001). Need for medical attention after vaccination was also significantly more frequent among the mRNA-1273 vaccinated group. Four participants reported a potential life-threatening reaction; however, the revision of their medical histories revealed that only one life-threatening report was correct (an autoimmune hepatitis reactivation after the second dose of the BNT162b2 vaccine, that was reported to the Spanish Pharmacovigilance Adverse Reactions System).

The frequency of adverse reactions after both the first and the second doses of both vaccine types was higher in participants who had a history of previous symptomatic COVID-19 infection (Table 3, Figure 1 and Figure 2). After a first dose of vaccine, 81.3% (95% CI 67.7–94.8%) of participants with a history of hospitalization for COVID-19 and 54.1% (95% CI 50.3–57.9%) of participants with a history of mild or moderate COVID-19 self-reported at least one adverse reaction, compared to 40.4% (95% CI 38.3–42.5%) of participants with no history of COVID-19 (*p* < 0.001). After a second dose of the vaccine, the frequency was 96.0% (95% CI 88.3–100%) for those with a history of hospitalization and 85.6% (95% CI 82.6–88.6%) for those with mild or moderate symptomatic COVID-19, significantly higher than for participants with no history or history of asymptomatic COVID-19 (*p* < 0.001). The ranking of the most common adverse effects was similar for the participants with no history of COVID-19 and the participants with a history of asymptomatic or mild to moderate COVID-19. However, for the participants who had suffered a hospitalization for COVID-19, fever and muscle or joint pain were ranked higher when ordered by frequency than other symptoms. Fever was the third most common symptom in the group that had been hospitalized for COVID-19 (62.5%, 95% CI 45.7–79.3% after dose 1) whereas it was the eighth most common symptom in other groups.

## 4. Discussion

This cross-sectional study on healthcare workers shows that the frequency of local and systemic adverse reactions was significantly higher in the mRNA-1273 vaccinated group compared with the BNT162b2 vaccinated group during the vaccination campaign in our centre. The BNT162b2 group had fewer reported occurrences than the mRNA-1273 group for all adverse reactions. A higher proportion of the mRNA-1273 group reported the need for medical attention or medical leave. Furthermore, our study shows that the seriousness of a previous COVID-19 infection can have an impact on the frequency of adverse reactions among vaccine recipients. Vaccine recipients with a history of symptomatic COVID-19 showed a significantly higher frequency of adverse reactions. Among participants with a history of hospitalization for COVID-19, fever was the third most common symptom after the first vaccine dose, a different pattern than that found in other groups where local adverse reactions were the most frequent. We found no significant differences in the frequency of adverse events when comparing vaccine recipients with no history of COVID-19 versus recipients with a history of asymptomatic COVID-19.

The frequency of adverse reactions by vaccine types in our study was similar to those reported in the clinical trials of BNT162b2 [5,14] and mRNA-1273 [7,15] vaccines when participants are stratified by similar age groups (data not shown). In the phase III clinical trial of the BNT162b2 vaccine, 84.7% of the vaccine recipients reported at least one local reaction and 77.4% one systemic reaction [5,14]. In the clinical trial of the mRNA-1273 vaccine, the majority of vaccine recipients reported local and systemic reactions, with higher rates after the second dose [7,15]. Our study showed a higher frequency of adverse reactions after the second dose both with the BNT162b2 and the mRNA-1273 vaccines, with a higher frequency for the mRNA-1273 group after both the first and second doses. Future studies should explore the biological aetiology of these differences in the two vaccines that use the same mRNA technology [16] and similar lipid nanoparticle delivery systems [17]. One possible explanation for the difference observed could be the amount of mRNA used in the vaccines. BNT162b2 [5] contains 30 μg of mRNA, while mRNA-1273 [7] contains 100 μg of mRNA. Additionally, mRNA-1273 has a higher concentration and dose (100 μg within 0.5 mL) compared to BNT162b2 (30 μg within 0.3 mL), which may also contribute to adverse reactions, especially injection site reactions.

Few studies have reported post-vaccination adverse events in healthcare workers. Among them, another cross-sectional study on healthcare workers in the USA reported similar frequencies and the same order of occurrence of local and systemic adverse events for both the BNT162b2 [18] and mRNA-1273 [19] vaccines. The authors published the results by type of vaccine in different articles, but they do not provide a comparison by type of vaccine. Likewise, a prospective survey of healthcare workers in South Korea compared adverse reactions following the first dose of the ChAdOx1 nCoV-19 Vaccine (AstraZeneca/Oxford) and the BNT162b2 vaccine [20]. The frequency of adverse reactions reported in this study after a first BNT162b2 dose was higher than in our study even if both study populations were similar in age and gender. Previous studies have also shown that a history of COVID-19 infection is associated with increased risk of adverse events [21,22,23,24,25]. Our study adds to other studies suggesting a dose-response relationship between the seriousness of a previous COVID-19 infection and adverse events to the COVID-19 vaccination.

This study has several limitations. Most of our study participants were female and younger than the general population, so these results cannot be generalized to other population groups. In the vaccination campaign at our centre, BNT162b2 was more available than mRNA-1273. Accordingly, our sample size for BNT162b2 was larger than for mRNA-1273, particularly for second vaccine doses. Participants in the mRNA-1273 group were more likely to report a history of COVID-19 infection than those in the BNT162b2 group. This difference may be explained because vaccination with mRNA-1273 started a month after vaccination with BNT162b2. During that month—January 2021—, Spain was suffering the third wave of COVID-19 with a high population incidence [26]. When the vaccination campaign started, there was no minimum interval between the COVID-19 infection and the administration of the vaccine. We did not register when the prior COVID-19 infection occurred, which could also be related to vaccination adverse effects. We did not obtain specific data about the severity of the local or systemic adverse reactions. However, we obtained information regarding the need for medical attention or medical leave. We may have overestimated or underestimated the frequency of adverse reactions since this is a cross-sectional study that used an ad hoc online self-reporting survey. We were able to review the medical histories of all the participants who reported a potential life-threatening reaction.

It is possible that those who suffered an adverse reaction were more predisposed to answer the questionnaire compared to those who did not suffer an adverse reaction. Nevertheless, the prevalences found in our study are similar to those of the aforementioned clinical trials. We had access to 50.6% participants’ medical records since it was optional to inform the health record ID code. Therefore, we were only able to review reported life-threatening reactions. Finally, the survey was available from 10 February 2021; therefore some answers were collected several days after the participants had been vaccinated. This limitation may have had a higher impact on the responses of those vaccinated with BNT162b2 vaccine, since it was the vaccine used during the first month of the campaign.

Excipients represent a major contributor to specific IgE and immediate allergic reactions associated with vaccines [27,28]. However, the aetiology of local or systemic adverse reactions to the COVD-19 vaccines is still unclear [19]. Our results point to a higher frequency of local and systemic adverse reactions in vaccine recipients who have a history of symptomatic COVID-19 and show that the seriousness of a previous COVID-19 infection could be associated with having more symptoms after vaccination. Therefore, it is possible that the local and systemic adverse reactions to COVID-19 vaccines that we have assessed are associated with the vaccines’ active ingredients and immune responses in addition to the excipients.

## 5. Conclusions

Widespread COVID-19 vaccination is a critical tool to help stop the pandemic [29]. After more than 5% of the world’s population has been vaccinated, it could be safe to say that the benefits of vaccination far outweigh the risks. So far, pharmacovigilance systems have observed that serious adverse reactions related with mRNA vaccines are very uncommon [30]. The effects recorded in our study, although frequent, are mild and well tolerated. However, studies regarding intention to get vaccinated against COVID-19 have shown increasing doubts about vaccine side effects and safety [31]. Adverse reactions to COVID-19 vaccines can cause fear and anxiety in the general population and may contribute to decreased willingness or delays in receiving a COVID-19 vaccine [32]. According to our results, adverse reactions were more frequently reported by the mRNA-1273 vaccinated group compared to the BNT162b2 vaccinated group. Moreover, vaccine recipients who had a history of COVID-19 infection reported more adverse reactions than those without a history of COVID-19, and the seriousness of the COVID-19 infection was associated with the frequency of adverse reactions. Our results could help inform vaccine recipients of their probability of having relatively common but mild adverse reactions to COVID-19 vaccines, increasing vaccine confidence and acceptance.

## Figures and Tables

**Figure 1 vaccines-09-01463-f001:**
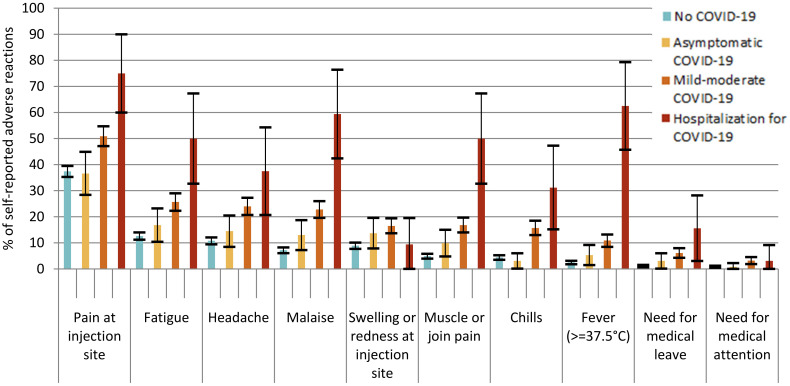
% of self-reported adverse reactions after first vaccine dose by history and seriousness of previous COVID-19 infection.

**Figure 2 vaccines-09-01463-f002:**
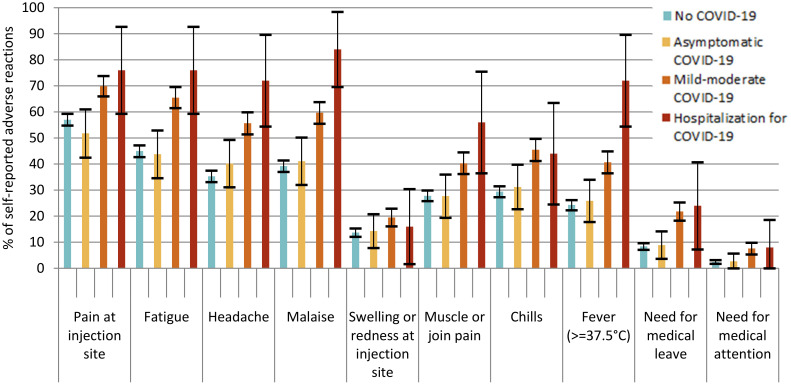
% of self-reported adverse reactions after second vaccine dose by history and seriousness of previous COVID-19 infection.

**Table 1 vaccines-09-01463-t001:** Sociodemographic characteristics and medical history of the study population.

		*n* = 2929	% (95% CI)
Gender			
	Women	2338	79.8% (78.4%, 81.3%)
	Men	589	20.1% (18.7%, 21.6%)
	Non-binary	2	0.1% (0%, 0.16%)
Age (in years)			
	Median (IQR)	34	33, 35
Age group			
	18–55	2251	76.9% (75.3%, 78.4%)
	>55	678	23.1% (21.6%, 24.7%)
Workers category			
	Medical doctor	586	20.0% (18.6%, 21.5%)
	Registered nurse	1386	47.3% (45.5%, 49.1%)
	Other, with patient contact	461	15.7% (14.4%, 17.1%)
	Other, without patient contact	496	16.9% (15.6%, 18.3%)
History of allergies			
	Yes	130	4.4% (3.69%, 5.18%)
	No	2799	95.6% (94.8%, 96.3%)
History of chronic illness			
	Yes	408	13.9% (12.7%, 15.2%)
	No	2521	86.1% (84.8%, 87.3%)
History of COVID-19 infection			
	Yes	817	27.9% (26.3%, 29.5%)
	No	2110	72.1% (70.5%, 73.7%)
Seriousness of COVID-19 infection			
	Asymptomatic	131	16.0% (13.5%, 18.6%)
	Mild or moderate symptoms	654	80.0% (77.3%, 82.8%)
	Hospitalization	32	3.9% (2.59%, 5.25%)
Vaccine type			
	BNT162b2 vaccine	2373	81% (79.6%, 82.4%)
	mRNA-1273 vaccine	506	17% (15.9%, 18.6%)
	Not reported	50	1.7% (1.24%, 2.18%)

**Table 2 vaccines-09-01463-t002:** Self-reported adverse reactions by vaccine type and dose.

		Dose 1				*p*-Value ^1^	Dose 2				*p*-Value ^1^
		BNT162b2		mRNA-1273			BNT162b2		mRNA-1273		
		*n* = 2373	% (95% CI)	*n* = 506	% (95% CI)		*n* = 2344	% (95% CI)	*n* = 121	% (95% CI)	
Occurrence of any adverse reaction		963	40.6% (38.6%, 42.6%)	303	59.9% (55.6%, 64.2%)	<0.001	1646	70.2% (68.4%, 72.1%)	111	91.7% (86.8%, 96.6%)	<0.001
Duration of the reaction (days, median, IQR)		3	2, 3	3	2, 4	<0.001	2	2, 3	3	2, 4	<0.001
	Pain at injection site	883	37.2% (35.3%, 39.2%)	292	57.7% (53.4%, 62.0%)	<0.001	1376	58.7% (56.7%, 60.7%)	99	81.8% (74.9%, 88.7%)	<0.001
	Fatigue	312	13.1% (11.8%, 14.5%)	155	30.6% (26.6%, 34.6%)	<0.001	1141	48.7% (46.7%, 50.7%)	86	71.1% (63.0%, 79.2%)	<0.001
	Headache	291	12.3% (10.9%, 13.6%)	118	23.3% (19.6%, 27.0%)	<0.001	935	39.9% (37.9%, 41.9%)	63	52.1% (43.2%, 61.0%)	0.008
	Malaise	213	9.0% (7.83%, 10.1%)	118	23.3% (19.6%, 27.0%)	<0.001	1008	43.0% (41.0%, 45.0%)	83	68.6% (60.3%, 76.9%)	<0.001
	Swelling or redness at injection site	192	8.1% (6.99%, 9.19%)	120	23.7% (20.0%, 27.4%)	<0.001	314	13.4% (12.0%, 14.8%)	55	45.5% (36.6%, 54.3%)	<0.001
	Muscle or joint pain	154	6.5% (5.50%, 7.48%)	86	17.0% (13.7%, 20.3%)	<0.001	713	30.4% (28.6%, 32.3%)	53	43.8% (35.0%, 52.6%)	0.002
	Chills	125	5.3% (4.37%, 6.17%)	83	16.4% (13.2%, 19.6%)	<0.001	744	31.7% (29.9%, 33.6%)	80	66.1% (57.7%, 74.5%)	<0.001
	Fever (≥37.5 °C)	82	3.5% (2.72%, 4.19%)	66	13.0% (10.1%, 16.0%)	<0.001	623	26.6% (24.8%, 28.4%)	80	66.1% (57.7%, 74.5%)	<0.001
	Nausea or vomiting	48	2.0% (1.46%, 2.59%)	41	8.1% (5.73%, 10.5%)	<0.001	275	11.7% (10.4%, 13.0%)	22	18.2% (11.3%, 25.1%)	0.034
	Insomnia	50	2.1% (1.53%, 2.68%)	13	2.6% (1.19%, 3.95%)	0.500	209	8.9% (7.76%, 10.1%)	17	14.0% (7.86%, 20.2%)	0.056
	Adenopathy	40	1.7% (1.17%, 2.20%)	19	3.8% (2.10%, 5.41%)	0.003	141	6.0% (5.05%, 6.98%)	12	9.9% (4.59%, 15.2%)	0.083
	Hives or rash	24	1.0% (0.61%, 1.41%)	19	3.8% (2.10%, 5.41%)	<0.001	44	1.9% (1.33%, 2.43%)	6	5.0% (1.09%, 8.83%)	0.033
Need for medical leave		37	1.6% (1.06%, 2.06%)	34	6.7% (4.54%, 8.90%)	<0.001	252	10.8% (9.50%, 12.0%)	31	25.6% (17.8%, 33.4%)	<0.001
Need for medical attention		27	1.1% (0.71%, 1.56%)	14	2.8% (1.34%, 4.20%)	0.120	76	3.2% (2.53%, 3.96%)	12	9.9% (4.59%, 15.2%)	0.010
Potential life-threatening reaction		0	0.0% (0%, 0%)	0	0.0% (0%, 0%)	>0.999	1	0.0% (0%, 0.13%)	0	0.0% (0%, 0%)	>0.999

^1^ Pearson’s Chi-squared test (for categorical variables) and Kruskal–Wallis rank sum test (for duration of the reaction).

**Table 3 vaccines-09-01463-t003:** Self-reported adverse reactions after first and second vaccine dose by history and seriousness of previous COVID-19 infection.

		No History of COVID-19		Asymptomatic COVID-19		Mild-Moderate COVID-19		Hospitalization for COVID-19		*p*-Value ^1^
		*n* = 2110	% (95% CI)	*n* = 131	% (95% CI)	*n* = 654	% (95% CI)	*n* = 32	% (95% CI)	
Vaccine dose 1										
	Any adverse reaction	852	40.4% (38.3%, 42.5%)	53	40.5% (32.1%, 48.9%)	354	54.1% (50.3%, 57.9%)	26	81.3% (67.7%, 94.8%)	<0.001
	Local adverse reaction	807	38.2% (36.2%, 40.3%)	48	36.6% (28.4%, 44.9%)	337	51.5% (47.7%, 55.4%)	24	75.0% (60.0%, 90.0%)	<0.001
	Systemic adverse reaction	462	21.9% (20.1%, 23.7%)	32	24.4% (17.1%, 31.8%)	261	39.9% (36.2%, 43.7%)	25	78.1% (63.8%, 92.4%)	<0.001
	Need for medical leave	23	1.1% (0.65%, 1.53%)	4	3.1% (0.11%, 6.00%)	40	6.1% (4.28%, 7.95%)	5	15.6% (3.04%, 28.2%)	<0.001
	Need for medical attention	18	0.9% (0.46%, 1.25%)	1	0.8% (0%, 2.25%)	21	3.2% (1.86%, 4.56%)	1	3.1% (0%, 9.15%)	<0.001
	Potential life-threatening reaction	0	0% (0%, 0%)	0	0% (0%, 0%)	0	0% (0%, 0%)	0	0% (0%, 0%)	>0.9
Vaccine dose 2		*n* = 1837	% (95% CI)	*n* = 112	% (95% CI)	*n* = 528	% (95% CI)	*n* = 25	% (95% CI)	
	Any adverse reaction	1233	67.1% (65.0%, 69.3%)	70	62.5% (53.5%, 71.5%)	452	85.6% (82.6%, 88.6%)	24	96.0% (88.3%, 100%)	<0.001
	Local adverse reaction	1062	57.8% (55.6%, 60.1%)	58	51.8% (42.5%, 61.0%)	376	71.2% (67.4%, 75.1%)	19	76.0% (59.3%, 92.7%)	<0.001
	Systemic adverse reaction	1127	6.4% (59.1%, 63.6%)	65	58.0% (48.9%, 67.2%)	437	82.8% (79.5%, 86.0%)	24	96.0% (88.3%, 100%)	<0.001
	Need for medical leave	154	8.4% (7.12%, 9.65%)	10	8.9% (3.65%, 14.2%)	115	21.8% (18.3%, 25.3%)	6	24.0% (7.26%, 40.7%)	<0.001
	Need for medical attention	45	2.4% (1.74%, 3.16%)	3	2.7% (0%, 5.67%)	40	7.6% (5.32%, 9.83%)	2	8.0% (0%, 18.6%)	<0.001
	Potential life-threatening reaction	0	0% (0%, 0%)	0	0% (0%, 0%)	1	0.2% (0%, 0.56%)	0	0% (0%, 0%)	0.3

^1^ Pearson’s Chi-squared test and Fisher’s exact test (for need for medical leave, need for medical attention and potential life-threatening reaction).

## Data Availability

The data presented in this study are available on request from the corresponding author. The data are not publicly available due to privacy restrictions.

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
