# Peer review of "Local and Systemic Adverse Reactions to mRNA COVID-19 Vaccines Comparing Two Vaccine Types and Occurrence of Previous COVID-19 Infection"

_vaccines, 2021, doi:10.3390/vaccines9121463_

Round 1

Reviewer 1 Report

In this manuscript entitled "Local and systemic adverse reactions to mRNA COVID-19 vaccines comparing two vaccine types and occurrence of previous COVID-19 infection," the author have adverse reaction to COVID-19 vaccines comparing the BNT162b2 or the mRNA-1273 COVID-19 vaccines and the presence and seriousness of a previous COVID-19 infection. I would like to see the manuscript in a concise and straightforward manner since the current version of the manuscript is large and covered several aspects. Overall, the study was planned and conducted in an appropriate way. The idea is interesting, the manuscript is written well, and the results were presented attractively. However, few issues need to be addressed before acceptance of this manuscript for publication.

  1. Abstract and Introduction are quite premature and shallow. Ideally, readers expect to have a very brief account of the aims, methods, key findings, and conclusions of a study from an abstract with a couple of sentences from each part. Besides, I can't find any discussion about both vaccine proteins as stated in the abstract.
  2. The authors didn't describe the methods in detail, which might be needed to ensure reproducibility of the results.
  3. Discussion should be improved in light of the results presented. Detailed discussion on predicted outputs and emerged variants needed to interpret and understand the consequences of variations. The
  4. The predictors' analysis is superficial and provides technical aspects of the resultant output. Author should provide critical justifications of the observed results.
  5. Additionally, I am confused about the discussion. It does not seem to really discuss the data that was described in the manuscript. I would suggest that the authors refocus their discussion to clarify how the results of their work fit into the larger picture of what is current today instead of describing more of a literature background.
  6. Page no 7, line no 167. Author reported ….Our study showed a higher frequency of adverse reactions after the second dose both  with  the  BNT162b2  and  the  mRNA-1273  vaccines,  although  higher for the mRNA-1273 group after both doses. I would like to know that which vaccine showed more side effect. What was the main side effect in the body, in particular age groups?
  7. Author should provide a comparison of both type of vaccine side effect in separate table
  8. Conclusion section missing in this Manuscript.

Author Response

Please see the attachment with the revised manuscript.

We thank Reviewer 1 for the review. We have tried to improve the paper according to your comments; however, we fear that we may not have understood all the comments. Some comments concern important parts of the paper (methods, discussion) but are very broad and unspecific. It has been difficult for us to understand what changes were needed. We would be happy to address any specific issues we may have missed in the first round of review.

R1: Abstract and Introduction are quite premature and shallow. Ideally, readers expect to have a very brief account of the aims, methods, key findings, and conclusions of a study from an abstract with a couple of sentences from each part. Besides, I can't find any discussion about both vaccine proteins as stated in the abstract.
A: Thank you for your comment. Regarding the abstract, we think it includes the aim of the study, a brief account of the methods used, the main findings and the conclusions. Unfortunately, we have reached the maximum words permitted in the abstract. Our abstract did not mention vaccine proteins, nor does the manuscript.
Regarding the Introduction, we feel that we have tried to keep it concise since there is a great amount of scientific information on COVID-19 vaccines at this moment. However, we would be glad to include any information that could help the readers of the paper. 

R1: The authors didn't describe the methods in detail, which might be needed to ensure reproducibility of the results.
A: In the Materials and Methods section we describe the population and sample of the study, the vaccination campaign, the type of study, the questionnaire and the variables, and the statistical analysis. If the reviewer feels any of this information needs improving, we will be glad to address them. We have included more information on the definition of allergies and chronic illnesses, including two references.

R1: Discussion should be improved in light of the results presented. Detailed discussion on predicted outputs and emerged variants needed to interpret and understand the consequences of variations. The
A: In the first part of the discussion, we summarize our results. In the second part, we compare our results to those of the phase III clinical trials and to other observational studies. In the third part, we state the main limitations of the study, and finally we explain the significance of our results. We have included some more information on the limitations of the study (highlighted in yellow) and we have explained better the differences in composition between BNT162b2 and mRNA-1273.
This was an observational cross-sectional study, we expected to have similar results to those of the clinical trials and that’s why we compare our results to those. We are not sure what the reviewer means by “emerged variants”. The scope of our study did not include a discussion on SARS-CoV-2 emerging variants of concern.

R1: The predictors' analysis is superficial and provides technical aspects of the resultant output. Author should provide critical justifications of the observed results.
A: We do not understand what the reviewer refers to as “predictors’ analysis” or “resultant output”. To the best of our knowledge, we did not perform a predictors’ analysis. We calculated the prevalence of adverse reaction comparing vaccine types and history and seriousness of previous COVID-19 infection. We think our results are in line with the objectives and the methods sections. Moreover, we think we provide an in depth discussion of the results in the discussion section. We will be happy to address any concern regarding the results of the paper.  

R1: Additionally, I am confused about the discussion. It does not seem to really discuss the data that was described in the manuscript. I would suggest that the authors refocus their discussion to clarify how the results of their work fit into the larger picture of what is current today, instead of describing more of a literature background.
A: In the first paragraph of the discussion, we summarize the results of the paper. In the second paragraph, we compare our results to those of the phase III clinical trials. In the third paragraph, we compare our results to similar observational studies (references 18-25). We think we are not offering a literature background, but comparing our results with pertinent studies. We have revised our references, and we feel they are pertinent for the discussion.

R1: Page no 7, line no 167. Author reported ….Our study showed a higher frequency of adverse reactions after the second dose both  with  the  BNT162b2  and  the  mRNA-1273  vaccines,  although  higher for the mRNA-1273 group after both doses. I would like to know that which vaccine showed more side effect. What was the main side effect in the body, in particular age groups?
A: The mRNA-1273 vaccine showed a higher prevalence of adverse reactions compared to BNT162b2 in both first and second dose. This result is explained in the Results section, in Table 2 and in the Discussion. We have rephrased the sentence of the Discussion section to: “Our study showed a higher frequency of adverse reactions after the second dose both with the BNT162b2 and the mRNA-1273 vaccines, although higher for the mRNA-1273 group after both first and second doses”.
Regarding the main side effects, we provide Figures 1 and 2 to answer that question. Unfortunately, as we state in our limitations our sample includes mainly young people, therefore we are not able to stratify both by age group and vaccine type. Moreover, our sample size for BNT162b2 is larger than for mRNA-1273. We have included this information in the Discussion “In the vaccination campaign of our center, BNT162b2 was more available than mRNA-1273. Accordingly, our sample size for BNT162b2 was larger than for mRNA-1273, particularly for second vaccine doses.”

R1: Author should provide a comparison of both type of vaccine side effect in separate table
A: In Table 2 we provide a comparison by vaccine type of the side effects. We need to include both vaccines in the same table since we compared the prevalence by vaccine type, and we offer a p value of the comparison. It is true that Table 2 is not well edited; we will contact the editor to know if we can present Tables 2 and 3 in a horizontal sheet.

R1: Conclusion section missing in this Manuscript.
A: The last paragraph of the discussion included the conclusions of the study. We have separated this paragraph with a new Conclusions heading.

Reviewer 2 Report

In this manuscript, Pares-Badell et al. compared the prevalence of AEs of two different types of Covid-19 mRNA vaccines, based on the data collected from healthcare workers in a hospital. They found that in general the mRNA-1273 vaccine exhibits higher occurrence of AEs after administration of 1st or 2nd doses than the BNT162b2 vaccine, and that those with previous Covid-19 history with symptoms or hospitalization showed higher rates of AEs than those with asymptomatic Covid-19 history or without Covid-19 history. However, no substantial difference was found between the group with history of allergies or chronic illnesses and the group without such history. The observations are in general consistent with, though the overall prevalence of AEs seems a little higher than, previous reports or clinical trails results.

The manuscript is well written, and the claims are overall supported by the data presented. It will help the society in clarifying the vaccine potential AEs and in encouraging people in selecting the types of vaccines. I thus would like to support the publication in Vaccines, if the following issues are addressed:

  • Please specify in tables 2 and 3, Which groups were compared to calculated the P values.
  • Please clarify that the in Table 2, Duration of the reaction, how the P-value was calculated (P<0.001), and whether a P value is needed here.
  • P7, line 173, Regarding the higher occurrence of AEs observed with Moderna vaccine,  higher total dose is most likely an explanation as the authors stated; however, Moderna vaccine also has a higher concentration (~0.2 ug/ul) than BNT162b2 (~0,1 ug/ul), this may contribute to the higher occurrence of AEs, especially injection site reactions. This possibility should be described.
  • In the discussion section, it needs to be mentioned that another limitation of the study is the imbalance of sample sizes between the two types of vaccines – Moderna group has much fewer responders than BNT. In addition, please also specify that the rates of completion of 2nd dose between Moderna (~24%?) and BNT162b2 (~98%?) groups are different.

Author Response

Please see the attachment with the revised manuscript. 

We thank Reviewer 2 for the kind comments and the thorough revision of the paper. We have included the changes proposed in the manuscript. Furthermore, we agree with the reviewer that the changes have improved the manuscript.

R2: Please specify in tables 2 and 3, Which groups were compared to calculated the P values.

A: In Table 2 we compare BNT62b2 versus mRNA1273. In Table 3 we compare the 4 categories of history and seriousness of previous COVID-19 infection. The tables had been edited by the journal and are difficult to read in the current format. We will contact the editor to try to make the tables more readable. However, we feel the tables are self-explanatory in terms of which groups are compared. We provide the p values and the confidence intervals. The confidence intervals can help assess which differences are significant when different categories are compared (for example, in table 3). We are not sure if we have answered to the reviewer's concerns, if it is not the case, we will be happy to consider this issue again.

R2: Please clarify that the in Table 2, Duration of the reaction, how the P-value was calculated (P<0.001), and whether a P value is needed here.

A: We have included in the caption of Table 2 and Table 3 information on which test was used for every variable. For Table 2: “Pearson's Chi-squared test (for categorical variables) and Kruskal-Wallis rank sum test (for duration of the reaction)”. For Table 3: “Pearson's Chi-squared test and Fisher's exact test (for need for medical leave, need for medical attention and potential life-threatening reaction).”
In the case of Duration of the reaction, since it is a continuous variable, we have used the Kruskal-Wallis rank sum test. We feel it is important to provide the p value since it indicates that the reaction after mRNA-1273 compared to BNT162b2 was significantly longer both for the first and the second doses. In the case of the first dose, even if the median is the same (2 days) the interquartile range is higher for mRNA-1273 and that’s why the difference can be (and is!) statistically significant.

R2: P7, line 173, Regarding the higher occurrence of AEs observed with Moderna vaccine,  higher total dose is most likely an explanation as the authors stated; however, Moderna vaccine also has a higher concentration (~0.2 ug/ul) than BNT162b2 (~0,1 ug/ul), this may contribute to the higher occurrence of AEs, especially injection site reactions. This possibility should be described.

A: We have added this information in the manuscript: “Additionally, mRNA-1273 has a higher concentration and dose (100 μg within 0,5 ml) compared to BNT162b2 (30 μg within 0,3 ml), which may also contribute to adverse reactions, especially injection site reactions.”

R2: In the discussion section, it needs to be mentioned that another limitation of the study is the imbalance of sample sizes between the two types of vaccines – Moderna group has much fewer responders than BNT. In addition, please also specify that the rates of completion of 2nd dose between Moderna (~24%?) and BNT162b2 (~98%?) groups are different.
A: We have included this information in the manuscript: “In the vaccination campaign of our center, BNT162b2 was more available than mRNA-1273. Accordingly, our sample size for BNT162b2 was larger than for mRNA-1273, particularly for second vaccine doses.”
However, in terms of the statistical analysis we think that an imbalance between the groups does not affect the results, since we had enough statistical power to assess small differences between the two groups.

Reviewer 3 Report

Dear Authors,

Your paper is quite a good read and very relevant in today's times where everyone wants to hear trends on the vaccines and their side effects. I have added some comments to improve the language, bring out highlights and critique the interpretations, all in the spirit of producing a good final product. Generally speaking, I need to be convinced that this data is statistically sound and applicable to a larger population. If not, I would like to see some disclaimers explaining the limited significance of the data.

For my specific comments, please see the PDF attached. I look forward to receiving a revised version of the manuscript. Thanks!

Reviewer

Author Response

Please see the attachment with the revised manuscript. 

We thank Reviewer 3 for this at length review. We appreciate the time and effort that you have dedicated to the manuscript. Your comments have been very valuable and we think they have helped improve the manuscript. We provide a point by point response. The changes have been highlighted within the manuscript.

R3: Suggest changing the title to something more informative, such as:
A: We are not able to see the suggested title. We would be happy to consider a more informative phrasing of the title.

R3: Please remove all author titles (MD/PhD/MPH) as this is inconsistent with other publications and delete 'and'?
A: We have removed the author titles and adapted the authors list to the Vaccines journal format.

R3: It is recommended to remove sub-section headings (background, results, conclusions, etc.) so that the abstract can be shorter and crisper.
A: We have removed the sub-section headings.

R3: For correct grammar, replace with ".... and history of COVID-19 infections". Replace 'to' with 'of' and add 'to'
A: We have made the suggested grammar changes in the abstract.

R3: Is it statistically accurate to compare two groups with such different n numbers (2373 versus 506)?
A: We think it is statistically accurate to make this comparison. We have calculated the statistical power to evaluate the differences between two proportions accepting an alpha risk of 0.05 and a beta risk of 0.2 in a two-sided test. With a sample of 2373 in one group and 506 in the other group we have enough statistical power to assess differences in the proportions of around 6.8%. Most of the differences we see in our study are over 7%. The fact that there is a ratio of 4.6 between both vaccine groups does not affect the statistical significance of the results. However, we have included this information in the discussion of the manuscript: “In the vaccination campaign of our center, BNT162b2 was more available than mRNA-1273. Accordingly, our sample size for BNT162b2 was larger than for mRNA-1273, particularly for second vaccine doses.”

R3: This is an important finding. Can be highlighted by saying, "Interestingly, ...."
A: We have included this suggestion in the abstract.

R3: Abstract conclusion: The statement “Our results could better help inform vaccine recipients of their probability of having adverse reactions to COVID-19 vaccines” is definitely not a conclusion of the analysis.
A: We understood the abstract conclusions as an interpretation of the significance of the results. Even if we agree that the statement is not a direct conclusion of the analysis, we feel it highlights the importance of our results and it can help the reader to interpret the significance of the findings.

R3: Please add more specific keywords.
A: We have added three keywords: COVID-19-vaccination; mRNA vaccines, healthcare workers.

R3: rephrase. Pandemic was declared for COVID-19 (disease caused by SARS-CoV-2)
A: We have rephrased the text to be more consistent with the WHO statement cited: “The World Health Organization characterized COVID-19 as a pandemic on March 11, 20201”.

R3: ECDC?
A: We have included the definition of the acronym (European Centre for Disease Prevention and Control).

R3: Replace 'in' with 'amongst'
A: we have included the change in the text.

R3: Please add precision and specify 'emergency-use authorization' and agency name (FDA/EMA).
A: We have specified the 'emergency-use authorization' and the agency name in the text. “The European Medicines Agency recommended the emergency use authorization of the BNT162b2 vaccine on 21 December 20209 and the mRNA-1273 vaccine on 6 January 202110.”

R3: This is a part of the Phase IV of any vaccine. Procedures are in place for adverse event reporting for both vaccines.
A: We agree with the reviewer that this is a part of the Phase IV of any vaccine. However, we feel it is also an important justification of our study, since we have studied the occurrence of mild adverse reactions to vaccines in healthcare workers.

R3: This suggests that all the healthworkers received their first dose on Feb 4. Please rephrase to clarify that the administration of the first doses was started on Feb 4.
A: We rephrased the sentence to “The administration of the first doses of the mRNA-1273 vaccine started on February 4, 2021”

R3: Please specify a reference. What medical guidelines define allergic reactions and chronic illnesses as listed?
A: We used the definition of allergic reactions of the Center for Disease Control (CDC) and we have added the reference. However, we had to adapt the CDC definition of chronic illnesses since there is a large degree of variation in the use of the term chronic disease within professional communities and the CDC definition omits chronic respiratory conditions (see Bernell 2016 “Use Your Words Carefully: What Is a Chronic Disease?” https://www.ncbi.nlm.nih.gov/pmc/articles/PMC4969287/)
We have added two references:
12. McNeil MM, DeStefano F. Vaccine-associated hypersensitivity. J Allergy Clin Immunol. 2018;141(2):463-472. doi:10.1016/J.JACI.2017.12.971
13. Centers for Disease Control and Prevention. About Chronic Diseases | CDC. Accessed November 18, 2021. https://www.cdc.gov/chronicdisease/about/index.htm

R3: Replace 'and' with 'while'
A: We have included this change in the manuscript.

R3: Specify units. i.e. Age in years
A: We have included this change in the manuscript.

R3: add 'that'
A: We have included this change in the manuscript.

R3: Was this adverse event reported to the vaccine manufacturer? If yes, this should be mentioned. It shows that the vaccine manufacturer's take responsibility and increases confidence amongst the readers/public.
A. The adverse event was reported to the Spanish Pharmacovigilance Adverse Reactions System. We have included this information in the manuscript.

R3: It is recommended to add gridlines to all the  tables as the numbers are not always aligned and may mislead. Please fix formatting of this row. The numbers are not aligned and confusing.
A: The journal has changed the formatting of the tables of the article. We agree with the reviewer that the new formatting needs to be fixed to improve readability. We will contact the editor regarding the table format.

R3: Replace with 'occurrence of any adverse reaction'
A: We have included this change in the manuscript.

R3: Please specify if this includes all participants or only those given one specific vaccine
A: It includes all participants, we have specified it is “after both the first and the second doses of both vaccine types”

R3: It is recommended that the authors provide a key / legend for the 4 colours used, and remove the related text from the X-axis. The figure will benefit from a clean-up.
R3: Same comment as Fig 1 regarding inclusion of a legend/key for the colours and clean-up of the X-axis.
A: We have added a legend for the 4 colours used and removed the related x-axis information. We agree with the reviewer, the figure looks much better now.

R3: Please elaborate. Across gender, age, co-morbidities....?
A: The word “cross-sectional” refers to the type of epidemiological study. In the first paragraph of the discussion we summarize the results and we feel it is useful to state the type of study.  

R3: replace with 'occurrence'
A: We have included this change in the manuscript.

R3: replace 'needed' with 'was reported to need'
A: We have included this change in the manuscript.

R3: Although this is an interesting conclusion, please add that the conclusion is from a relatively small dataset and remains to be confirmed for a larger population.
A: We feel that it is better to report all the limitations of the study in the 4th paragraph of the discussion and avoid separating the discussion of specific limitations in different paragraphs. However, we feel our sample size is not small from the statistical point of view. We feel the main limitation of our sample lies in its’ characteristics: healthcare workers who are younger and have a high female to male ratio. However, we have included in the discussion a sentence highlighting the imbalance between the two vaccine groups, particularly for the second doses of mRNA-1273 vaccine: “In the vaccination campaign of our center, BNT162b2 was more available than mRNA-1273. Accordingly, our sample size for BNT162b2 was larger than for mRNA-1273, particularly for second vaccine doses.”

R3: Please add citations/references to the papers where this information was published.
A: We have added the citations.

R3: This always makes the reader suspicious. Please consider including this data.
A: To compare our results with the results of the clinical trials was not one of our objectives; however, we feel in the context of the discussion it is useful to let the reader know our results are similar to those of the clinical trials. We can include an annex with our results stratified by age groups, but we feel it would be out of the scope of our objectives. Furthermore, even if we are able to stratify our sample by age, our sample may differ with the samples of the clinical trials in other characteristics that we cannot control.

R3: The delivery systems are not identical. Please re-phrase.
A: Both vaccines use lipid nanoparticle delivery systems but with different components. We were not aware of this fact. We have rephrased the sentence to “two vaccines that use the same mRNA technology and similar lipid nanoparticle delivery systems”. We have included a reference were the reader can find the components of the nanoparticle lipids.

R3: Edit for better grammar
A: We have rephrased the sentence to “BNT162b25 contains 30 μg of mRNA dose, while mRNA-12737 contains 100 μg.”

R3: Is this because people who got vaccinated a month prior, may have forgotten to report? Please clarify in the text.
A: We do not know why participants vaccinated with mRNA-1273 were more likely to report a history of COVID-19. However, we think it could be because mRNA-1273 vaccination started a month after BNT162b2 and during that month (January 2021) the population incidence of COVID-19 was high in Spain since we were suffering the third wave of COVID-19. We have added this information and a reference in the manuscript. 

R3: Recommend deleting this statement as it has been said before.
A: We have deleted the statement.

R3: Not clear. Please clarify.
A: We have clarified the statement “It is possible that those who suffered an adverse reaction were more predisposed to answer the questionnaire compared to those who did not suffer an adverse reaction”

R3: In addition to the excipients. It is well documented and recognized that the excipients lead to side-effects / adverse effects.
A: We have included this change in the manuscript.

R3: replace 'is' with 'could be'
A: We have included this change in the manuscript.

R3: replace with 'recorded'
A: We have included this change in the manuscript.

R3: Please correct grammar. Sentence misleading.
A: We have changed the sentence to “Moreover, vaccine recipients who had a history of COVID-19 infection reported more adverse reactions than those without a history of COVID-19”

R3: Please correct grammar.
A: We have corrected the sentence. “Our results could help inform vaccine recipients of their probability of having relatively common but mild adverse reactions to COVID-19 vaccines, increasing vaccine confidence and acceptance.”

R3: It is recommended to merge graphs in Annex 2 (one dose) and Annex 3 (two doses) for efficient comparisons. It is also recommended to add a legend/key to specify use of colour and clean-up X-Axis by removing the vaccine name.
A: We have added the key to the graphs in Annex 2. However, we feel it is better not to merge Annex 2 and Annex 3 to keep the same format we have used in Figures 1 and 2, separating vaccination dose 1 and dose 2.

Round 2

Reviewer 1 Report

We have assessed the findings presented here and referred to the previous literature. In the context of the data presented in the current paper improve to our previous expectations with regards to originality and we are therefore able to consider it for publication as present form.